# NeuroBE: Escalating Neural Network Approximations of Bucket Elimination

**Sakshi Agarwal**[1]   **Kalev Kask**[1]   **Alexander Ihler**[1]   **Rina Dechter**[1]

[1]University of California Irvine

## Abstract

A major limiting factor in graphical model inference is the complexity of computing the partition function. Exact message-passing algorithms such as *Bucket Elimination (BE)* require exponential memory to compute the partition function; therefore, approximations are necessary. In this paper, we build upon a recently introduced methodology called *Deep Bucket Elimination (DBE)* that uses classical Neural Networks to approximate messages generated by *BE* for large buckets. The main feature of our new scheme, renamed *NeuroBE*, is that it customizes the architecture of the neural networks, their learning process and in particular, adapts the loss function to the internal form or distribution of messages. Our experiments demonstrate significant improvements in accuracy and time compared with the earlier *DBE* scheme.

## 1 INTRODUCTION

Two of the critical goals of probabilistic modeling are the compact representation of probability distributions and the efficient computation of their marginals and modes. Probabilistic graphical models, such as Markov networks [Pearl, 1988, Darwiche, 2009, Dechter, 2013] provide a framework to represent distributions compactly as normalized products or factors : $P(X) = \frac{1}{Z} \prod_\alpha f_\alpha(X_\alpha)$, where $X$ is a set of variables, each potential $f_\alpha$ is a function over a subset $X_\alpha$ of the variables (its scope) and $Z = \sum_X \prod_\alpha f_\alpha(X_\alpha)$ is the *partition function*. Computing the partition function is exponential in the induced width of the model's graph even for distributions that admit a compact representation.

The partition function $Z$ is defined by two types of operations: sums and products. It can be evaluated efficiently if $\sum_X \prod_\alpha f_\alpha(X_\alpha)$ can be reorganized using the distributive law along a variable ordering [Dechter, 2003]. This

organization can be described using buckets as data structures, one for each variable in the ordering. When a bucket is processed, its associated variable is removed, creating a bucket output function, also called a *message*, that is passed to a subsequent bucket. The time and space complexity of computing this function is exponential in its number of arguments, called scope or the bucket's width. Overall, Bucket Elimination (*BE*) [Dechter, 1999b] is time and memory exponential in the induced-width of the model's graph along the ordering.

Providing good approximations to *BE* is important not only because it generates an answer to a query, but primarily because it compiles a structure and a set of messages that can be used to answer multiple queries (e.g., the probability of evidence for various evidence variables Darwiche [2009]). Also, the messages can be used as building blocks for generating heuristics for search to further improve performance. We therefore consider and evaluate *NeuroBE* in the context of *approximate BE*, generating approximation to its messages.

Schemes that approximate *BE* include (weighted) mini-bucket (*WMB*) [Dechter and Rish, 2003, Liu and Ihler, 2012] and generalized belief propagation schemes [Yedidia et al., 2000, Mateescu et al., 2010]. A recently introduced scheme, *Deep Bucket Elimination (DBE)* [Razeghi et al., 2021] approximates each bucket function with a neural network (NN). While this approach is inherently time consuming, requiring the independent training of many NNs to solve the partition function of a single problem, it has yielded more accurate approximations on several benchmarks when compared against competing schemes. Both *WMB* and *DBE* are restricted by memory. Yet the memory demanded by *WMB* (notwithstanding recent work [Forouzan and Ihler, 2015]) increases exponentially with its $i$-bound not accommodating refined steps of memory increase. In contrast, NN architectures can grow more gradually and may facilitate a more flexible memory-accuracy balance.

*Accepted for the 38th Conference on Uncertainty in Artificial Intelligence* (UAI 2022).

**Contributions.** We present *NeuroBE*, a re-design of *DBE*, that addresses the shortcomings of its *one size fits all* policy by customizing the NN construction and training sample size to each bucket separately, in proportion to its message size. We also introduce a new loss function that is sensitive to a bucket's message distribution, also called *local structure*. We also provide an analysis relating the local errors to an upper bound on the global error. In an extensive empirical evaluation we show that *NeuroBE* outperforms *DBE* across all benchmarks using far less resources, such as training samples and NN size, while yielding higher accuracy with less time. We provided the source code to reproduce the results of this paper at https://github.com/dechterlab/NeuroBE.

The paper is organized as follows. We first provide a background to *BE*, *WMB* and *DBE*; then we present *NeuroBE*; followed by error analysis; lastly, we demonstrate the efficiency of *NeuroBE* empirically.

**Related work.** As noted, approximating and bounding *Bucket Elimination* has been carried out extensively over the years for all probabilistic queries. Well known is the *Mini-Bucket Elimination* scheme [Dechter and Rish, 2003] and its variants, such as *Weighted Mini-Bucket Elimination (WMB)*, augmented with message-passing cost-shifting [Liu and Ihler, 2011b].

Neural network approximation to *BE* was introduced in Razeghi et al. [2021]. The idea is closest in spirit to the Neuro-Dynamic Programming scheme as outlined in Bertsekas and Tsitsiklis [1996] where the cost-to-go functions (similar to messages) generated by dynamic programming can be approximated by NNs. This technique is also highly related to Deep Reinforcement Learning (DRL) [Mnih et al., 2015] where, in the absence of a model, the value function is approximated by NNs learned from temporal trajectories.

Recently, *Graph Neural Networks (GNNs)* [Scarselli et al., 2009] are used to learn *messages* following the message-passing reasoning methods in graphical models [Abboud et al., 2020, Yoon et al., 2018, Heess et al., 2013]. However, Yoon et al. [2018], Heess et al. [2013] is restricted to small instances (i.e., ~40 variables) and Abboud et al. [2020] tackles problems with a known polynomial-time approximation. GNN based methods derive a supervised end-to-end learning algorithm, generalizing across different problem instances. In contrast, we consider a different class of algorithms, where we confine learning to within a problem instance *only*.

## 2 BACKGROUND

A graphical model can be defined by a 3-tuple $\mathcal{M} = (\mathbf{X}, \mathbf{D}, \mathbf{F})$, where $\mathbf{X} = \{X_i : i \in V, V = \{1, ..., n\}\}$ is a set of $n$ variables indexed by $V$, and $\mathbf{D} = \{D_i : i \in V\}$ is the set of finite domains for each $X_i$ (i.e. each $X_i$ can only

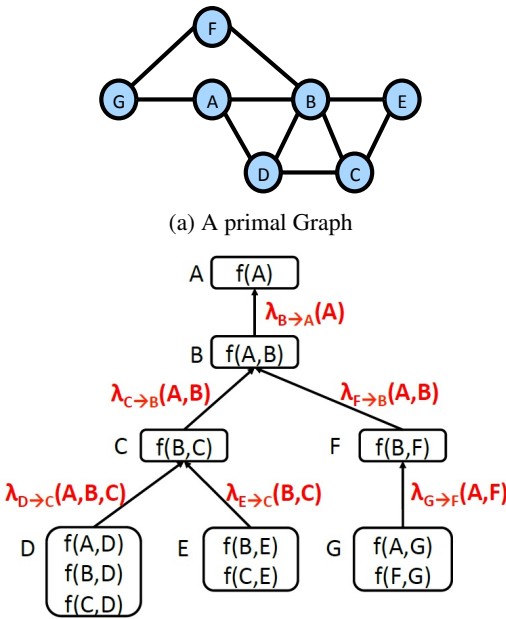

(a) A primal Graph

(b) Bucket Elimination example

Figure 1: (a) A primal graph of a GM with 7 variables. (b) Illustration of *BE* with an ordering A B C E D F G.

assume values in $D_i$, and each $D_i$ is finite). Each function $f_\alpha \in \mathbf{F}$ is defined over a subset of the variables called its scope, $X_\alpha$, where $\alpha \subseteq V$ are the indices of variables in its scope, and $D_\alpha$ denotes the Cartesian product of their domains so that $f_\alpha : D_\alpha \to R \geq 0$.

The **primal graph** of a graphical model associates each variable with a node. An edge between node $i$ and node $j$ is created if and only if there is a function containing $X_i$ and $X_j$ in its scope. Figure 1a shows a primal graph of a graphical model with variables indexed from $A$ to $G$ and functions over pairs of variables are connected by an edge. Graphical models can be used to represent a global function, often a probability distribution, defined by $Pr(X) \propto \prod_\alpha f_\alpha(X_\alpha)$. An important task is to compute the normalizing constant, also known as the partition function $Z = \sum_X \prod_\alpha f_\alpha(X_\alpha)$.

### 2.1 BUCKET ELIMINATION

*Bucket Elimination (BE)* [Dechter, 1999a] is a universal exact algorithm for probabilistic inference. It is a variable elimination algorithm that can answer a wide-range of queries, including the partition function ranging from constraint satisfaction, to pure combinatorial optimization (e.g., Most Probable Explanation (MPE/MAP)), and weighted counting (Partition Function, Probability of Evidence).

Given a variable ordering $d$, BE (presented in Algorithm 1, omitting steps 9-12) creates a *bucket tree* where each node is a bucket representing a variable in the ordering $d$. Figure 1b shows a bucket tree for the primal graph in Figure

1a along an ordering. Each bucket in this tree contains a set of the model's functions depending on the given order of processing. For example, Bucket G in Figure 1b has functions $\{f(A, G), f(F, G)\}$, an exhaustive set of model's functions with variable G in its scope. There is an arc from a bucket, say $B_c$, to a parent bucket, $B_p$, if $X_p$ is the latest variable in bucket $B_c$'s message scope along the ordering (constants are placed in $B_1$). In the same example, there is an arc from Bucket G to Bucket F.

*BE* performs inference along the bucket tree as a 1-iteration message-passing algorithm (bottom-up). It processes each bucket from leaves to the root passing messages from child ($c$) to parent ($p$). For a child variable $X_c$, *BE* encompasses all the functions in bucket $B_c$. This includes the original functions in the graphical model as well as the messages received by processing previous variables. It then marginalizes $X_c$ out from the product of functions in $B_c$ generating a new, so called, *bucket function* or message, denoted $\lambda_{c\to p}$, or $\lambda_c$ for short:

$$\lambda_c = \sum_{X_c} \prod_{f_\alpha \in B_c} f_\alpha \qquad (1)$$

The $\lambda_c$ function is placed in $B_p$, the bucket of $X_p$. Once all the variables are processed, *BE* outputs all the messages and the exact value of Z by taking the product of all the constants present in the bucket of the first variable. We illustrate *BE* message flow in our example problem in Figure 1b.

**Complexity.** Both the time and space complexity of *BE* are exponential in the **induced width**, which is the size of the largest number of variables in the scope of any message over all buckets [Dechter, 2013]. Clearly, *BE* becomes impractical if the induced width is large.

## 2.2 WEIGHTED MINI-BUCKET

Given a variable ordering $d$, *Weighted Mini-Bucket* (*WMB*) [Dechter and Rish, 2003, Liu and Ihler, 2011b] approximates *BE* by partitioning each bucket $B_c$ with high width into several disjoint "mini-buckets" $B_c^j$ to ensure that individual $B_c^j$ has low ($\le i-$bound) width. The method also assigns a weight $p_{cj}$ to each mini-bucket $B_c^j$. *WMB* then eliminates the bucket's variable $X$ in the $j^{th}$ mini-bucket $B_c^j$ using the power sum following Holder's inequality [Liu and Ihler, 2011a]:

$$\mu_c^j = \Big( \sum_X \prod_{f_\alpha \in B_c^j} f_\alpha^{\frac{1}{p_{cj}}} \Big)^{p_{cj}},$$

and $\mu^j{}_c$ is passed to a parent bucket $B_p$. For example, using an $i$-bound = 2 in Figure 1b, *WMB* approximates the exact message $\lambda_{D\to C}(A, B, C)$, passed from bucket $D$ to bucket $C$, by three messages corresponding to partitioning bucket $B_D$ into three mini-buckets each with a single function

$f(A, D), f(B, D), f(C, D)$. Based on Holder's inequality [Liu and Ihler, 2011a], the exact message is bounded by the product of the mini-bucket messages when the weights $p_{cj}$'s are non-negative and sum to one. Thus, for any $i$-bound *WMB* generates an upper bound of the partition function.

Generally, time and accuracy in *WMB* increases with the $i$-bound. Yet, due to memory constraints it can run with a maximum $i$-bound of about 20 and therefore, the generated bounds can be extremely loose when a problem's induced-width is high. Interestingly, when it is run, *WMB* terminates quickly, taking a few seconds and up to a minute.

## 2.3 DEEP BUCKET ELIMINATION

Given a variable ordering $d$, *Deep Bucket Elimination (DBE)* approximates each message generated in the bucket tree whenever the scope ($S$) of a message is high ($> i$-bound) using a neural network (NN). Following the previous example of $i$-bound = 2 in Figure 1b, rather than sending the exact message from bucket $D$ to bucket $C$, *DBE* sends a NN $\mu_{\theta,D\to C}(A, B, C)$ parameterized by $\theta$ that approximates the exact message $\lambda_{D\to C}(A, B, C)$, as we elaborate next.

We use $\mu_{c\to p}^*$ to denote the *local exact message* computed using all functions in bucket $c$, regardless of the local functions being exact or approximate (as defined by the right side of Eq. (1)). However, if we execute exact *BE*, in which case the bucket contains exact messages only, we denote the output message as $\lambda_{c\to p}$ and refer to it as the *global exact message*.

Let $B$ be a bucket with width $w > i$-bound and $\mu^*(S)$ be its local exact message having scope $S$ whose size is the bucket's width, $w$. *DBE* constructs a fully-connected feed-forward NN having $w$ nodes in the input layer, followed by $L$ hidden layers each having $h$ hidden nodes with $ReLU$ activation function. The output layer contains one node with a real-valued output. Subsequently, *DBE* generates a training set $\{(s_n, \mu^*(s_n))\}$ of size N, where $s_n$ is the $n^{th}$ configuration of $S$, sampled uniformly at random and where $\mu^*(s_n)$ is the local exact message value defined in Eq. (1). The NN function $\mu_\theta(S)$ approximating $\mu^*(S)$ is trained to minimize the mean square error loss :

$$L(\theta) = \frac{1}{N} \sum_{n=1}^{N} \big( \mu^*(s_n) - \mu_\theta(s_n) \big)^2.$$

Once training is complete, *DBE* passes the trained NN, $\mu_\theta$ to its parent bucket.

While *DBE* showed superior quality of solutions compared with *WMB*, its time performance was quite inferior. In particular, training each bucket message used the same fixed architecture and the same sample size (quite large), needlessly resulting in a high total time. This paper is devoted

to a redesign of the algorithm, aiming to improve both time and accuracy, as we elaborate in the following section.

# 3 NEUROBE

Algorithm *NeuroBE* advances *DBE*, and is described in this section, focused on the partition function task. However, extension to other queries is straightforward, and requires altering only the message definition in Eq. (1) to fit the corresponding task (e.g., replacing summation by maximization in Eq. (1) for the MAP query.)

We rename *DBE* to *NeuroBE* since we use mostly shallow neural networks (up to 2 layers). Algorithm 1 describes *NeuroBE*. The algorithm first creates a bucket tree along a given ordering (line 2). It then processes buckets one by one along the ordering from last to first. If the current processed bucket has width $w \leq i$-bound, then the message, $\mu_{c \to p}^*$ is computed exactly (line 7). Otherwise, the bucket's message is approximated by a neural net (line 9). The message is placed in the appropriate parent bucket in the bucket tree. Finally, line 13 calculates the partition function using the functions in bucket $B_1$. Note that if a bucket contains a NN function, then computing $\mu^*$ (line 7 or 9) requires evaluating the trained NN (see Algorithm 3, line 4).

The difference between *NeuroBE* and *DBE* is solely in the individual message approximation scheme, *NN-train*. In contrast to *DBE*, *NeuroBE* dynamically customizes the NN architecture and training set size to the bucket's message complexity, and it modifies the loss function to depend on the message distribution. These modifications are described in the sequel.

---

**Algorithm 1** NeuroBE

**Input**: Graphical model $\mathcal{M} = (\mathbf{X}, \mathbf{D}, \mathbf{F})$, Ordering $d = X_1, ..., X_n$
**Parameters**: $i$-bound $i$; #layers $L$; constants $b, \eta$;
**Output**: the partition function constant and bucket messages

1: **for** c in n...1 **do**
2:    (Initialize buckets) put all unplaced functions mentioning $X_c$ in $B_c$.
3: **end for**
4: **for** c in n...1 **do**
5:    Let $X_p$ be the parent variable of $X_c$ in the bucket-tree
6:    **if** $width(B_c) < i$ **then**
7:       compute $\mu_{c \to p}^* \leftarrow \sum_{X_c} \prod_{f_\alpha \in B_c} f_\alpha$,
8:    **else**
9:       $\mu_{\theta, c \to p} \leftarrow$ *NN-train*$(\{f_\alpha | f_\alpha \in B_c\}, L, b, \eta)$
10:    **end if**
11:    Put $\mu_{c \to p}^*$ or $\mu_{\theta, c \to p}$ in $B_p$
12: **end for**
13: $\hat{Z} = \sum_{X_0} \prod_{f_\alpha \in B_1} f_\alpha$
14: **return** $\hat{Z}$ and all messages generated

---

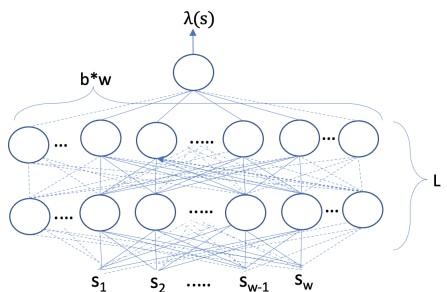

Figure 2: For a bucket of width $w$, we illustrate a NN architecture with $L (= 2)$ layers and $b \cdot w$ hidden-units with $b \geq 1$.

**NN Architecture selection.** Clearly, the NN size should depend on both the approximated function's complexity and, especially, its dimensionality. Since a bucket message's scope size is the induced-width, $w$, we make the number of hidden units, $h$, a function of $w$ while keeping the number of layers, $L$, constant. Specifically, we use a simple function $h = b \cdot w$, where $b \geq 1$ to fit the NN's architecture to the message size. Figure 2 is an example NN model architecture with an input layer of size $w$ and 2 hidden layers with dimension $h$. Next, we provide a rule to determine sample sizes to train a NN, depending on a notion of its complexity.

**NN complexity.** The notion of a *Pseudo-dimension* [Pollard, 1984, Anthony and Bartlett, 2002] is often used to measure the expressive power of a set of functions that can be learned by any statistical regression algorithm. The work in Bartlett et al. [2019] derived lower bounds to the pseudo-dimension for NNs with ReLU activation function (an architecture used in our work). We use the derived lower bound to estimate the pseudo-dimension ($\rho$) of a NN ($\mu_\theta$), having an architecture of $L$ layers and $b \cdot w$ hidden units, yielding (see Appendix for derivation):

$$\rho \propto (L * b * w)^2 \log(b * w). \tag{2}$$

Since in our experiments, the pair $(L, b)$ are fixed for a given problem instance, our $\rho$ estimate only varies with $w$ and is used to determine the sample complexity.

**Sample Complexity.** As suggested in Vapnik [1999], we choose a sample size for training a NN, $\mu_\theta$, proportional to its pseudo-dimension, Eq. (2). We therefore select a number of samples $N$ satisfying the expression

$$N = \eta * (L * b * w)^2 \log(b * w), \tag{3}$$

where $\eta$ is a constant allowing us to tweak $N$ linearly. Since the triplet $(L, b, \eta)$ of a problem is fixed, the number of samples for training, $N$, is a function of $w$ only.

**Sample Generation** Let $B$ be a generic bucket where variable $X$ is eliminated; let $F$ be the set of functions from

**Algorithm 2** generate-samples($X, F, N$)

**Input**: $X$, a variable to be eliminated, $F$, a set of functions over scope $S \cup \{X\}$, $N$, an integer,
**Output**: $\mathcal{D}$, a set of $N$ samples

1: initialize $\mathcal{D} = \{\}$,
2: **for** $i = 1..N$ **do**
3:     $s \leftarrow$ sample uniformly from domain($S$)
4:     $\mu^*(s) \leftarrow \sum_x \prod_{f \in F} f(s, x)$     {Eq. (1)}
5:     Add $(s, \mu^*(s))$ to $\mathcal{D}$
6:     Update $\mu^*_{min}, \mu^*_{max}$
7: **end for**
8: Normalize $\mathcal{D}$ (Eq 4)
9: **return** $\mathcal{D}$

---

the graphical model (initialized in line 2, Algorithm 1) as well as messages from the previous buckets (line 11, Algorithm 1) residing in $B$ and $S$ be the scope of the output message function $\mu^*$. Then, Algorithm 2 generates a dataset $\mathcal{D}$ containing a given number of samples $N$. The algorithm iteratively and uniformly at random, samples a configuration $\{S = s\}$ from the domain of $S$ and computes the exact local bucket function value for $s$ using Eq 1 (lines 3,4). The pair $<s, \mu^*(s)>$ is added to the dataset $\mathcal{D}$ (line 5). A normalization step occurs in line 10, where each sample $s$ is shifted and scaled to the range $[-1, 1]$ and $\mu^*(s)$ is shifted and scaled to $[0, 1]$, to accelerate training of the NN [Le Cun et al., 1991], by:

$$\mu^*_{norm}(s) = \frac{\mu^*(s) - \mu^*_{min}}{\mu^*_{max} - \mu^*_{min}} \qquad (4)$$

where $\mu^*_{min}, \mu^*_{max}$ (line 6) are defined relative to the dataset $\mathcal{D}$ by $\mu^*_{min} = \min_{s \in D} \mu^*(s)$ and $\mu^*_{max} = \max_{s \in D} \mu^*(s)$.

**Loss Function** Algorithm *DBE* sampled each message input configuration uniformly and uses the mean square error loss function for training. However, it seems intuitive that generating the samples by taking into account the message distribution could lead to more effective training of the function. Since sampling directly from the message distribution is hard, we instead weight each sample by an *importance weight* within the loss function, described next.

**Definition 1** (I.m.s.e loss). Let $\mu_\theta$ be the NN for approximating the function $\mu^*_{norm}$. Let $D = \mathcal{D}_{Train}$ be the training set. Then, the I.m.s.e loss function for a given mini-batch, $\mathcal{D}_i \in \mathcal{D}$ of size $\#\mathcal{D}_i$ is defined by:

$$L_{\mathcal{D}_i}(\mu^*_{norm}, \mu_\theta) = \frac{1}{\#\mathcal{D}_i} \sum_{s \in \mathcal{D}_i} (\mu^*_{norm}(s) - \mu_\theta(s))^2 * W(s),$$
$$(5)$$

where

$$W(s) = \frac{\mu^*(s)}{\sum_{s' \in \mathcal{D}_{Train}} \mu^*(s')}. \qquad (6)$$

**Log transformations** Usually in our experiments we apply a log transformation to the input functions, for computational reasons. The algorithms presented here remain the same; however the values $\mu^*$, $\mu^*_{min}$ and $\mu^*_{max}$ in this case refer to the $\log$ of the original function values. In cases when we use the log-space computation, the weight function $W(s)$ (Eq. 6) is not suitable. We instead use modified importance weights,

$$W^{\log}(s) = \frac{\log \mu^*(s) - \log \mu^*_{min}}{\sum_{s' \in \mathcal{D}_{Train}} (\log \mu^*(s') - \log \mu^*_{min})} \qquad (7)$$

Note that the importance weight, $W(s)$ or $W^{\log}(s)$, are computed in the original function space that is not normalized.

**MaskedNet** For problems with determinism, i.e., a high proportion of zero probability states, a fully connected feed-forward NN was unable to correctly predict deterministic outputs and hence Razeghi et al. [2021] used a MaskedNet. The input configuration is sent to a fully connected layer with a RELU activation function to obtain a feature vector. This feature vector is then sent to two sister layers: the first layer outputs a binary mask responsible for determining whether the final output is zero, and the second layer is responsible for predicting the target value of the Bucket's function. The activation functions of the two final layers are the logistic function and the softplus function, respectively. The outputs from the two sister networks are multiplied together to get the final output of the MaskedNet. The loss for the MaskedNet in *NeuroBE* is thus a sum of the binary cross-entropy loss (from the first output layer) and the proposed I.m.s.e loss (from the second output layer). Thus when a sample configuration $s$ has $\mu^*(s) = 0$, the loss becomes the binary cross-entropy error, since $W(s) = 0$, following Eq. (5) and (6).

**NN-Train** Algorithm 3 describes the procedure *NN-Train*. Its input parameters are $L, b, \eta$ where $L$ is the number of layers, $b$ is a constant to determine the number of hidden units, $b \cdot w$ (line 1), and $\eta$ is another constant to determine the training sample size $N$ (line 2, Eq. (3)). A major step occurs next where the algorithm generates a dataset $\mathcal{D}$ and splits it into the training set $\mathcal{D}_{Train}$ of size $N$, validation set $\mathcal{D}_{Val}$ of size $N/4$ and testing set $\mathcal{D}_{Test}$ of fixed size (50k) (lines 3-4; see also Algorithm 2). Lines 8-12 then describe the batch training for updating the NN parameters $\theta$ using the I.m.s.e loss function (line 11, Eq. (5)), and the Adam optimizer [Kingma and Ba, 2014] (line 12) with a learning rate of 0.001 and a batch-size of 256 across all benchmarks. At the end of each epoch, the current model is evaluated on a holdout validation set (line 14). We evaluate the early-stopping criteria (line 15), which is assigned *True* when either the maximum limit $\#epochs$ is reached

**Algorithm 3** NN-train($F, X, L, b, \eta, \#epochs$)

**Input**: $F$, a set of functions over scope $S \cup \{X\}$ where $X$ is to be removed, $w$ scope size.
**Parameters**: $L$: # layers in NN, $\#epochs$, $\eta$, $b$: constants
**Output**: $\mu_\theta$: NN message approximation, $\hat{\epsilon}$: an estimated bucket error bound

1: $\#h \leftarrow b * w$
2: $N \leftarrow \#$ training samples$(w, \eta, L, b)$ {Eq. 3}
3: $\mathcal{D} \leftarrow$ generate-samples$(X, F, N + N/4 + 50k)$
4: $D_{Train}, D_{Val}, D_{Test} \leftarrow$ Split$(\mathcal{D})$
5: Initialize NN parameters $\theta$, $p$=1, early-stopping $\leftarrow$ False
6: **while** p $\leq \#epochs$ and $\neg$ early-stopping **do**
7:    $D_1, .., D_k \leftarrow$ divide $D_{Train}$ to minibatch
8:    **for** $i = 1..k$ **do**
9:       Let $D_i = \{(s, \mu^*_{norm})\}$
10:       Compute $\{\mu_\theta(s) | s \in D_i\}$
11:       $loss_{D_i} \leftarrow L_{\mathcal{D}_i}(\mu^*_{norm}, \mu_\theta)$ {Eq. 5}
12:       $\theta \leftarrow$ update $\theta$ by optimize(Adam, $loss_{D_i}, \theta$)
13:    **end for**
14:    $loss_{D_{val}} \leftarrow L_{D_{val}}(\mu^*_{norm}, \mu_\theta)$ {For stop condition}
15:    early-stopping $\leftarrow$ evaluate early-stopping$(loss_{D_{Val}})$
16:    $p \leftarrow p + 1$
17: **end while**
18: Unnormalize $\{\mu_\theta(s), \mu^*(s) | s \in D_{Test}\}$ {Inverse of Eq. 4}
19: $\hat{\epsilon} \leftarrow \max_{s \in D_{Test}}(log\mu^*(s) - log\mu_\theta(s))$
20: **return** $\mu_\theta, \hat{\epsilon}$

---

or the validation error increases for two consecutive epochs. Once training is complete, we compute the maximum log relative error between the target and NN approximated messages over a test set (lines 18-19). In the next section, we use this maximum error to analyse the propagation of error in *NeuroBE*. The *NN-train* procedure then returns the approximated message $\mu_\theta$, along with its estimated error.

**Complexity.** The time and space complexity for learning a single message in *NeuroBE* is a function of the NN and sample size. In contrast to *DBE*, here the NN and sample sizes vary with the bucket's width.

## 4 ERROR ANALYSIS

We now analyse the relationship between the local errors contributed by each approximated message and the global partition function error, focusing on a simple case where the bucket tree is a chain.

**Definition 2** (local and global bucket errors). Given a bucket $B$, let $\lambda$ be the (global) exact message generated in $B$, $\mu^*$ be the (local) exact message in $B$ at the time of message computation, and $\mu = NN\text{-}train(\mu^*)$ be its NN approximation. Then, we define the local and global log relative errors

as:

$$E = \log \mu^* - \log \mu,$$

and,

$$G = \log \lambda - \log \mu.$$

We use log relative error since it simplifies the analysis. We now show the following relationship:

**Theorem 1.** *Assume a bucket-chain along an ordering $d$, and let $B_c$ be a bucket along the chain at position $c$ having scope $S$ of its bucket message. Let $E_c(s) = \log \mu^*_c(s) - \log \mu_c(s)$ and let $\epsilon_c = \max_{s \in D(S)} |E_c(s)|$. Then,*

$$G_c = \log \lambda_c - \log \mu_c \leq \sum_{k=0}^{n-c} \epsilon_{c+k}$$

*In particular, since $\lambda_1 = Z$ and $\mu_1 = \hat{Z}$,*

$$G_1 = \log Z - \log \hat{Z} \leq \sum_{k=0}^{n-1} \epsilon_{1+k} \quad (8)$$

For the proof see the supplementary material.

## 5 EMPIRICAL EVALUATION

### 5.1 EXPERIMENTAL SETUP

We conducted experiments comparing *NeuroBE* against *WMB* [Dechter and Rish, 2003, Liu and Ihler, 2012] and *DBE* [Razeghi et al., 2021] over several benchmarks. We also compare the impact of the two loss functions, m.s.e and l.m.s.e, on the performance of *NeuroBE*. Finally, we illustrate how increasing sample and NN complexity impact performance.

**i-bounds.** All three algorithms, *WMB*, *DBE* and *NeuroBE*, use the i-bound parameter ($i$). As noted, in *WMB* higher i-bounds lead to more accurate bounds with more time and memory, up to their memory limit. Algorithms *DBE* and *NeuroBE* are also observed to improve accuracy and time with increasing i-bounds because of the reduced number of trained buckets $\#NB(i)$. Hence, for a fair comparison we use an $i$-bound of 10 for some (easy) benchmarks, while primarily using the highest feasible $i$-bound of 20 dictated by *WMB*'s memory bound for other (hard) benchmarks.

**Benchmarks** Following the example of *DBE*, we evaluated *NeuroBE* on instances selected from three well-known benchmarks from the UAI repository used in Kask et al. [2020]: grids (vision domain), pedigree (genetic linkage analysis) and DBNs. We targeted diverse benchmarks (in structure and level of determinism) and aimed for different levels of hardness. Thus, in the grids benchmark, we distinguish those problems that can be solved exactly, which we

**(a) pedigree**

| Problem Description (i-bound=20) | | | | | ref logZ | WMB error | #NB | DBE (#h=100, N=320K) statistics on error (5 runs) | | | | NeuroBE (#h=3w, Nmin=49K) Statistics on resources | | | Loss: mean square error statistics on error (5 runs) | | | | Loss: Imp. mean square error statistics on error (5 runs) | | | |
|---|---|---|---|---|---|---|---|---|---|---|---|---|---|---|---|---|---|---|---|---|---|---|
| Id | name | k | #v | w | | error | | avg error | min error | stdev | time (h) | hmax | Navg | Nmax | avg error | min error | std | time (h) | avg error | min error | std | time (h) |
| 1 | pedigree13 | 3 | 888 | 33 | -31.18 | 6.4696 | 127 | 5.32 | 2.62 | 3.06 | 11 | 96 | 218K | 706K | 9.04 | 7.86 | 0.8 | 7.3 | 1.11 | 0.76 | 0.25 | 7.5 |
| 2 | pedigree41 | 5 | 885 | 32 | -76.04 | 4.1497 | 92 | 4.27 | 3.25 | 0.73 | 9.9 | 93 | 190K | 658K | 10.2 | 8.4 | 2.16 | 5.6 | 0.47 | 0.153 | 0.21 | 6.2 |
| 3 | pedigree51 | 5 | 871 | 35 | -77.27 | 9.7624 | 120 | 23.92 | 9.23 | 12.7 | 13 | 102 | 259K | 809K | 11.73 | 10.37 | 1.26 | 10.4 | 3.51 | 1.96 | 0.89 | 10.4 |
| 4 | pedigree34 | 5 | 922 | 33 | -64.23 | 7.0762 | 106 | 5.91 | 1.57 | 5.98 | 11 | 96 | 211K | 706K | 6.14 | 4.56 | 4.14 | 6.5 | 0.65 | 0.23 | 0.29 | 6.96 |
| 5 | pedigree7 | 4 | 867 | 34 | -64.82 | 6.0012 | 108 | 11.26 | 5.18 | 7.8 | 11 | 99 | 350K | 900K | # | # | # | 10.7 | 1.75 | 1.21 | 0.7 | 10.7 |
| 7 | pedigree19 | 5 | 693 | 28 | -59.020 | 2.5809 | 43 | 6.054 | 5.41 | 0.92 | 9.4 | 71 | 149K | 482K | 9.14 | 8.7 | 0.6 | 4.6 | 2.61 | 1.91 | 0.6 | 5.2 |

(a) pedigree

**(b) Grid-hard**

| Problem Description (i-bound=20) | | | | | ref logZ | WMB error | #NB | DBE (#h=100, N=320K) statistics on error (5 runs) | | | | NeuroBE (#h=w, Nmin=19K) Statistics on resources | | | Loss: mean square error statistics on error (5 runs) | | | | Loss: Imp. mean square error statistics on error (5 runs) | | | |
|---|---|---|---|---|---|---|---|---|---|---|---|---|---|---|---|---|---|---|---|---|---|---|
| Id | name | k | #v | w | | error | | avg error | min error | stdev | time (h) | hmax | Navg | Nmax | avg error | min error | std | time (h) | avg error | min error | std | time (h) |
| 1 | grid4040f10 | 2 | 1600 | 55 | 5490 | 215.45 | 308 | 97.1 | 11.8 | 65.2 | 12 | 55 | 120K | 364K | 14.57 | 0.15 | 8.94 | 7.69 | 9.71 | 2.1 | 0.91 | 6.4 |
| 2 | grid4040f5 | 2 | 1600 | 55 | 2800 | 84.92 | 308 | 39.9 | 6.28 | 35 | 12 | 55 | 120K | 364K | 8.86 | 2.05 | 8.21 | 7.66 | 3.7 | 0.2 | 3.06 | 6.3 |
| 3 | grid4040f2 | 2 | 1600 | 55 | 1220 | 25.24 | 308 | 7.34 | 1.2 | 5.4 | 12 | 55 | 120K | 364K | 3.15 | 1.48 | 1.66 | 7.48 | 2.28 | 1.2 | 0.91 | 6.1 |
| 4 | grid4040f15 | 2 | 1600 | 55 | 8200 | 338.2 | 308 | 83.46 | 41.8 | 34.2 | 13 | 55 | 75K | 228K | 24.75 | 6.62 | 18.1 | 5.7 | 17.87 | 7.45 | 9.16 | 5.7 |
| 5 | grid4040f10w | 2 | 1600 | 114 | 5637 | 297.7 | 376 | 100.5 | 6.4 | 82.2 | 21 | 114 | 150K | 670K | 67.76 | 32.7 | 26 | 11 | 54.18 | 25.01 | 21.1 | 11.1 |
| 6 | grid4040f5w | 2 | 1600 | 114 | 2819 | 136.99 | 376 | 78.2 | 72.6 | 5.6 | 21 | 114 | 150K | 670K | 5.37 | 1.65 | 4.77 | 11.6 | 9.62 | 4.62 | 5.27 | 12.1 |
| 7 | grid4040f2w | 2 | 1600 | 114 | 1231 | 32 | 376 | 15.12 | 0.92 | 20.5 | 18 | 114 | 150K | 670K | 10.15 | 8.81 | 1.44 | 11.8 | 5.56 | 2.92 | 2.11 | 10 |

(b) Grid-hard

**(c) Grid-easy**

| Problem Description (i-bound=10) | | | | | ref logZ | WMB error | #NB | DBE (#h=100, N=320K) statistics on error (5 runs) | | | | NeuroBE (#h=w, Nmin=20K) Statistics on resources | | | Loss: mean square error statistics on error (5 runs) | | | | Loss: Imp. mean square error statistics on error (5 runs) | | | |
|---|---|---|---|---|---|---|---|---|---|---|---|---|---|---|---|---|---|---|---|---|---|---|
| Id | name | k | #v | w | | error | | avg error | min error | stdev | time (h) | hmax | Navg | Nmax | avg error | min error | std | time (h) | avg error | min error | std | time (h) |
| 1 | grid1010f10w | 2 | 100 | 21 | 333.32 | 32 | 31 | 4.45 | 0.89 | 3.71 | 1.5 | 21 | 51K | 87K | 1.58 | 0.28 | 0.75 | 0.35 | 1.83 | 0.98 | 0.76 | 0.48 |
| 2 | grid1010f10 | 2 | 100 | 13 | 303.09 | 1.58 | 8 | 0.7 | 0.05 | 0.54 | 0.3 | 13 | 23K | 30K | 1.28 | 0.59 | 0.41 | 0.06 | 1.18 | 0.86 | 0.32 | 0.05 |
| 3 | grid2020f2 | 2 | 400 | 27 | 291.73 | 11.24 | 114 | 1.98 | 0.36 | 1.28 | 5.3 | 27 | 68K | 177K | 0.4 | 0.013 | 0.36 | 2.24 | 0.13 | 0.015 | 0.14 | 2.5 |
| 4 | grid2020f10 | 2 | 400 | 27 | 1312 | 80.86 | 114 | 10.04 | 1.05 | 9.3 | 5.5 | 27 | 68K | 177K | 2.51 | 1.62 | 1.31 | 2.22 | 2.4 | 0.37 | 1.98 | 2.27 |
| 5 | grid2020f5 | 2 | 400 | 27 | 665.12 | 39.44 | 114 | 5.75 | 0.24 | 3.1 | 5.5 | 27 | 68K | 177K | 0.84 | 0.37 | 0.53 | 1.95 | 0.8 | 0.081 | 0.78 | 2.3 |
| 6 | grid2020f15 | 2 | 400 | 27 | 1963 | 122.91 | 114 | 17.8 | 3.08 | 9.8 | 3 | 27 | 68K | 177K | 2.39 | 0.41 | 2.16 | 2.2 | 2.68 | 1.08 | 2.54 | 2.2 |

(c) Grid-easy

**(d) DBN**

| Problem Description | | | | | ref logZ | WMB error | #NB | DBE (#h=100, N=320K) statistics on error (5 runs) | | | | NeuroBE Statistics on resources | | | Loss: mean square error statistics on error (5 runs) | | | | Loss: Imp. mean square error statistics on error (5 runs) | | | |
|---|---|---|---|---|---|---|---|---|---|---|---|---|---|---|---|---|---|---|---|---|---|---|
| Id | name | k | #v | w | | error | | avg error | min error | stdev | time (h) | hmax | Navg | Nmax | avg error | min error | std | time (h) | avg error | min error | std | time (h) |
| **i-bound=20** | | | | | | | | | | | | **NeuroBE (#h=3w, Nmin=147K)** | | | | | | | | | | |
| 1 | rbm20 | 2 | 40 | 21 | 58.53 | 0.0007 | 20 | 0.22 | 0.03 | 0.17 | 1.1 | 60 | 147K | 147K | 0.37 | 0.08 | 0.29 | 0.45 | 0.37 | 0.06 | 0.25 | 1.73 |
| 2 | rbm21 | 2 | 42 | 22 | 63.15 | 6.39 | 22 | 0.48 | 0.27 | 0.19 | 1.3 | 63 | 163K | 164K | 0.9 | 0.59 | 0.2 | 0.46 | 0.57 | 0.15 | 0.26 | 1.75 |
| 3 | rbm22 | 2 | 40 | 21 | 66.55 | 8.65 | 24 | 0.47 | 0.14 | 0.35 | 1 | 66 | 180K | 182K | 0.59 | 0.02 | 0.38 | 0.57 | 0.75 | 0.48 | 0.36 | 1.76 |
| 4 | rbm-ferro20 | 2 | 44 | 23 | 151.16 | 0.005 | 20 | 1.33 | 0.29 | 1.21 | 1 | 60 | 147K | 147K | 1.11 | 0.38 | 0.75 | 0.44 | 0.75 | 0.37 | 0.32 | 1.13 |
| 5 | rbm-ferro21 | 2 | 42 | 22 | 152.62 | 1.98 | 22 | 3.43 | 0.83 | 1.89 | 1.2 | 63 | 163K | 164K | 2.3 | 1.27 | 1.08 | 0.48 | 1.82 | 1.06 | 0.87 | 2.5 |
| 6 | rbm-ferro22 | 2 | 44 | 23 | 166.11 | 0.517 | 24 | 6.52 | 3.86 | 1.5 | 1.3 | 66 | 180K | 182K | 5.32 | 2.515 | 2.69 | 0.59 | 4.17 | 3.8 | 0.32 | 1.29 |
| **i-bound=10** | | | | | | | | | | | | **NeuroBE (#h=3w, Nmin=82K)** | | | | | | | | | | |
| 1 | rbm20 | 2 | 40 | 21 | 58.53 | 7.85 | 30 | 0.49 | 0.11 | 0.35 | 1.6 | 100 | 82K | 87K | 1.57 | 0.89 | 1.04 | 0.3 | 1.43 | 0.83 | 0.82 | 0.8 |
| 2 | rbm21 | 2 | 42 | 22 | 63.15 | 15.73 | 32 | 0.73 | 0.14 | 0.47 | 1.7 | 105 | 90K | 109K | 1.62 | 0.68 | 0.98 | 0.4 | 0.89 | 0.51 | 0.42 | 1.02 |
| 3 | rbm22 | 2 | 40 | 21 | 66.55 | 27.46 | 34 | 0.65 | 0.19 | 0.41 | 1.8 | 110 | 99.5K | 121K | 0.89 | 0.03 | 0.6 | 0.46 | 0.94 | 0.83 | 0.61 | 1.3 |

(d) DBN

Figure 3: Results on performance of *NeuroBE* against *DBE* and *WMB*. $k$: domain size, $\#v$: variables, $w$: induced width, $\#NB$: number of buckets that are trained with NNs, $\#h$: number of hidden units per layer (reported maximum $\#h$ for *NeuroBE*), $N$: number of training samples (reported minimum, average and maximum $\#N$ for *NeuroBE*), $error$: L1 error for referenced and estimated $\log Z$ (reported minimum, average, and standard deviation over 5 runs for *DBE* and *NeuroBE*), $time$: average time taken to get the estimated $error$, $\#$ in a cell denotes estimated partition function is $-\infty$. *Note: Here, reference $\log Z$ is approximated by Kask et al. [2020]

call "grid-easy", from those that cannot be solved, called "grid-hard". We also distinguish benchmarks that possess *determinism*, namely have a high proportion of zero probabilities, since it can impact training. We randomly selected 13 instances from Grids, with easy ones (400 variables, width 20-30) and hard ones (1600 variables, width 55 or 114), 6 from pedigrees ($\approx$800 variables, width $\approx$34), which posses high level of determinism and 6 from DBNs ($\approx$40 variables, width $\approx$22), totalling 25 instances. As described in section 3, we apply $\log$ transformations to Grids and DBNs since they have large message function values.

**NN architectures and sample sizes.** Through a process of trial and error on a selected instance from each benchmark, we selected the parameters of the architectures and sample sizes as follows. We selected $L = 1, h = 3w$, and $N_{avg} \in [149k, 350k]$ for pedigrees; $L = 2, h = 3w$, and $N_{avg} \in [80k, 180k]$ for DBN; $L = 2, h = w$, and $N_{avg} \in [23k, 68k]$, for grid-easy; and $L = 1, h = w$, and $N_{avg} \in [75k, 150k]$ for grid-hard.

**Performance measures** We evaluate the performance of *NeuroBE* using: $error = |\log_e Z - \log_e \hat{Z}|$ where $\hat{Z}$ estimates $Z$. When the exact $Z$ is not available (i.e., for hard Grid benchmark), we use $Z^*$ as a surrogate to $Z$, which is obtained using an advanced sampling scheme for a duration of $100 * 1hr$ [Kask et al., 2020].

## 5.2 RESULTS

Figure 3 compares *NeuroBE* against *WMB* and *DBE* over the four benchmarks. The first few columns show the problem statistics for instances in the respective benchmarks (pedigree, grid-hard; grid-easy and DBN). We then show results on *WMB*'s error, followed by *DBE*'s and *NeuroBE*'s performance information. We omit *WMB*'s time performance in Figure 3 since its execution takes only a minute. For *DBE*, we report the number of buckets trained by NNs, ($\#NB$), followed by the average error, minimum error, standard deviation, and average time (in hours) over five runs, which is preferable due to stochasticity. For *NeuroBE* with both m.s.e and I.m.s.e loss functions, we also report statistics about NN architecture and sample size that varies within problem instances: the average and maximum number of training samples, $N_{avg}$ and $N_{max}$, and maximum number of hidden units, $h_{max}$.

**Pedigrees** We observe immediately that, overall, *NeuroBE* with the I.m.s.e loss function is clearly superior to *DBE* and *NeuroBE* with m.s.e loss. In particular, it is $\geq 5$ times more accurate than *DBE* for almost all instances and takes less time, since it uses far less training samples. *NeuroBE* with the I.m.s.e loss function also outperforms *WMB* on most instances. *NeuroBE* with the m.s.e loss function is less accurate than both *WMB* and *DBE* for most instances. It also fails to approximate the partition function for instance

| i-bound=20 | | | h=3*w | | | | h=5*w | | | |
|---|---|---|---|---|---|---|---|---|---|---|
| Id | #v | w | #N$_{avg}$ (10$^3$) | standard deviation | error | t(h) | #N$_{avg}$ (10$^3$) | standard deviation | error | t(h) |
| 1 | 888 | 33 | 218 | 0.25 | 1.11 | 7.5 | 437 | 0.17 | 0.34 | 14.3 |
| 3 | 871 | 35 | 259 | 0.89 | 3.51 | 10.4 | 518 | 1.57 | 1.42 | 18.3 |
| 7 | 693 | 28 | 149 | 0.6 | 2.61 | 5 | 298 | 0.51 | 0.68 | 8.2 |

(a) pedigree

| i-bound=20 | | | h=w | | | | h=w | | | |
|---|---|---|---|---|---|---|---|---|---|---|
| Id | #v | w | #N$_{avg}$ (10$^3$) | standard deviation | error | t(h) | #N$_{avg}$ (10$^3$) | standard deviation | error | t(h) |
| 5 | 1600 | 114 | 120 | 10.82 | 55.11 | 12 | 262 | 16.3 | 17.76 | 17 |
| 7 | 1600 | 114 | 120 | 3.6 | 5.5 | 10 | 262 | 1.39 | 2.34 | 14.5 |

(b) Grid-hard

| i-bound=10 | | | h=w | | | | h=3*w | | | |
|---|---|---|---|---|---|---|---|---|---|---|
| Id | #v | w | #N$_{avg}$ (10$^3$) | standard deviation | error | t(h) | #N$_{avg}$ (10$^3$) | standard deviation | error | t(h) |
| 1 | 100 | 21 | 51 | 0.76 | 1.83 | 0.48 | 202 | 0.08 | 1.2 | 1.13 |
| 2 | 100 | 13 | 23 | 0.32 | 1.18 | 0.05 | 38 | 0.15 | 0.12 | 0.09 |

(c) Grid-easy

| i-bound=20 | | | h=3*w | | | | h=5*w | | | |
|---|---|---|---|---|---|---|---|---|---|---|
| Id | #v | w | #N$_{avg}$ (10$^3$) | standard deviation | error | t(h) | #N$_{avg}$ (10$^3$) | standard deviation | error | t(h) |
| 1 | 40 | 21 | 147 | 0.25 | 0.37 | 1.73 | 245 | 0.07 | 0.2 | 3.2 |
| 2 | 42 | 22 | 163 | 0.26 | 0.57 | 1.75 | 272 | 0.24 | 0.77 | 3.7 |
| 3 | 40 | 21 | 163 | 0.36 | 0.75 | 1.76 | 300 | 0.22 | 0.51 | 4.5 |
| 4 | 44 | 23 | 147 | 0.32 | 0.75 | 1.13 | 245 | 0.61 | 0.67 | 2.17 |
| 5 | 42 | 22 | 164 | 0.87 | 1.82 | 1.3 | 272 | 0.23 | 1.06 | 2.6 |
| 6 | 44 | 23 | 182 | 0.32 | 4.17 | 1.29 | 300 | 0.86 | 3.43 | 2.9 |

(d) DBN

Figure 4: Performance of NeuroBE when increasing # sample &/or NN complexity. $N_{avg}$: average samples, $t(h)$: average time, $Error$: global error (reported average and standard deviation over 5 runs)

5 (errors and standard deviation denoted by # in Figure 3). Here, *DBE* has similar or even worse accuracy than *WMB*.

**Grid-hard.** The results for the grid-hard benchmark is shown in Fig. 4b. We used the highest possible $i$-bound of 20, and we observe that *NeuroBE* can achieve a far lower average error and standard deviation and takes far less time than *WMB* and *DBE*, particularly with the I.m.s.e loss. In most cases, we see a reduction in time by a factor of two. *DBE* outperforms *WMB* across all problem instances.

**Grid-easy.** The results for the grid-easy benchmark is shown in Figure 4c. We used a lower $i$-bound of 10 to facilitate the training of a relatively large number of buckets. As expected, when an instance has a low induced-width and only a small number of buckets are approximated (e.g., ID 2), *DBE* obtains high accuracy. As the induced-width increases and more buckets are trained, *NeuroBE* has far higher accuracy compared with both *WMB* and *DBE*. Both the loss functions in *NeuroBE* show similar performance.

**DBN** We report results for the DBN benhcmark for two $i$-bounds. Overall, the results are mixed. For $i$-bound = 20, *NeuroBE* achieves a higher accuracy than *DBE* for half of the instances with far less #training samples (but with more training time). It is superior to *WMB* on instances 2, 3, and 5. When comparing the two loss functions in *NeuroBE*, I.m.s.e loss has better (or similar) performance for most instances. However, *WMB* performs better on instance 1, 4, and 6, as the induced-width is closer to the $i$-bound. For $i$-bound = 10, *DBE* and *NeuroBE* show better accuracy than *WMB* for those three instances. *DBE* has better accuracy when compared with *NeuroBE*, using more #training samples (and hence, more time). *NeuroBE* with I.m.s.e loss is better performing compared with m.s.e loss on most instances. Overall, *NeuroBE* when trained with I.m.s.e loss takes more time than with m.s.e loss for the same #training samples.

In summary, *NeuroBE* using I.m.s.e compared against *DBE* is about 50% faster while also far more accurate on pedigrees, twice as fast and 5 to 10 fold more accurate on hard grids. It is also faster and more accurate on easy grids and has a mixed but still comparable performance on DBNs.

**The impact of loss functions.** We observe that *NeuroBE* with the I.m.s.e loss shows better performance (lower average error and standard deviation) than *NeuroBE* with the m.s.e loss for the pedigree instances and the majority of DBN, grid-easy and grid-hard instances. An F-test with a significance level of $0.05$ on the two groups of partition function estimates (each consisting of five approximations) showed that the means are significantly different for pedigrees, in Figure 3(a). For the grids and DBN, there was no statistical difference between the two means. However, by inspection, we see a reduction in the standard deviation for almost all instances.

**Impact of architecture size.** Figure 4 shows the impact of architecture size on time and accuracy for a few problem instances. We show results for two different NN architectures and their associated sample sizes. As expected, we see that increasing the sample and NN sizes increases both time and accuracy for pedigrees. For grid-hard instances, we just increased the sample sizes and kept the same architecture having $h = w$. We observe that the average error is reduced, as expected. Instances from grid-easy and DBN (except ID 2) show a similar improvement in performance with a larger NN and training sample size. This trend illustrates that increasing the size of the NN (matched by a suitable increase in sample size), improves the accuracy of *NeuroBE*, at the cost of more time and memory. A key question for future work is how to develop a policy that can facilitate gradual control of architecture and sample size increase to improve performance in an anytime way.

# 6 CONCLUSION & FUTURE WORK

In this work, we advance the earlier theme of using Neural Networks to approximate the class of bucket-elimination algorithms that is at the heart to probabilistic reasoning. *NeuroBE* can be viewed as a realization of Neuro-Dynamic Programming schemes [Bertsekas and Tsitsiklis, 1996], in the context of graphical models. That being said, it requires the training of numerous NNs per problem instance, and thus, the central aim of *NeuroBE*'s design (customizing NN architectures, training samples, and the loss function to the message) is to enhance *efficiency* and *scalability* of such schemes. We presented *NeuroBE* and illustrated on challenging instances over three benchmarks that it can be far more accurate and requires less time compared with *Deep Bucket Elimination* (*DBE*). It is also superior to *weighted mini-bucket* (*WMB*) even when provided with the highest memory resources feasible.

**Future Work.** We will explore further how to improve *NeuroBE*'s efficiency by customizing additional features of a NN and its training per bucket (e.g., varying the number of layers). We will also explore moving from training buckets separately per single variable to training clusters of buckets within a tree-decomposition, thus training a single function per union of buckets [Dechter, 2013], potentially reducing the number of trained functions at the cost of more time for sample generation. Finally, we will explore parameter sharing by training multiple bucket functions simultaneously in a single problem instance and across a benchmark of instances.

# 7 ACKNOWLEDGEMENT

We thank our reviewers and our lab colleagues Bobak Pezeshki and Nick Cohen for their constructive feedback. This work was supported in part by NSF grants IIS-2008516.

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
