# OpenReview forum: "NeuroBE: Escalating NN Approximations of Bucket Elimination"
_auai.org/UAI/2022/Conference — UAI 2022 Poster_

### Official Review · Reviewer_jRvg · 2022-03-23

**Q2(1) Originality/Novelty:** 2
**Q2(2) Significance/Impact:** 2
**Q2(3) Correctness/Technical Quality:** 3
**Q2(6) Clarity Of Writing:** 3
**Q6 Overall Score:** 7
**Q8 Confidence In Your Score:** 3

**Q1 Summary And Contributions:**

The existing work, Deep Bucket Elimination (DBE), replaced each large message (a function) in the Bucket Elimination algorithm with a NN to improve memory usage. This paper improved the NN architecture, and training dataset size N, dynamically adapting them to the message size (width $w$) rather than having it constant. They proposed a new loss function, analysed the NN complexity, error bounds, and evaluated the approach showing clear improvements over DBE in terms of runtime and accuracy.

**Q2 Assessment Of The Paper:**

More detailed information regarding each of these aspects is given below:

**Q2(4) Quality Of Experiments (Optional):**

4: Excellent: The experimental evaluation is comprehensive and the results are compelling.

**Q2(5) Reproducibility:**

3: Good: Key resources (e.g., proofs, code, data) are available and key details (e.g., proofs, experimental setup) are sufficiently well-described for competent researchers to confidently reproduce the main results.

**Q3 Main Strengths:**

The presented results appear promising, a clear improvement over the previous work in DBE. The ideas being explored are interesting.

**Q4 Main Weakness:**

The paper appears to be relatively incremental work. The many different symbols (and inconsistent use of them) make it harder to read.

**Q5 Detailed Comments To The Authors:**

# Questions

Q1: The paper mentions scaling input features to $[-1,1]$ in case the domain size $k > 2$, and this is the case for pedigree. I wonder why not use a standard one-hot encoding? Is this about ordinal variables rather than nominal (and was this the case for pedigree)?

Q2: What are the consequences of having the samples uniform at random? The DBE paper included future work: "We generated samples uniformly at random while other policies should be considered, for example sampling in proportion to the target function distribution" which was seemingly left out in this paper while other future work is present. Any reason why this would not be of interest?

Q3: How different are the values in $\mu^*$ or $\lambda$, in the used benchmarks? Is there a lot of structure - most obvious one being all same values -- that would make it easier for the NN to represent the correct function?

Q4: What is the value of $\eta$ in the experiments?

Q5: "Squared error on message values that are large seem to be potentially much costlier than smaller message values because of their significant contribution to the sum-product operation in Eq 1". I do not understand this, can you elaborate on this motivation? Let's say we have a bucket $f(S) \times f(S') \times \dots$, where $f(S)$ was approximated in a previous bucket as $\mu (s)$ and $|\mu^{ * }(s) - \mu(s)| = x$ for both $s=1$ and $s=2$. Does the impact of error $x$ not only depend on what it will be multiplied with? If both $\mu(s_1)$ and $\mu(s_2)$ are eventually multiplied with the same values (e.g. $y$) then the absolute error is $x * y$ for both and hence the impact is the same, regardless of whether $\mu^{ * }(s_1)$ is larger than $\mu^{ * }(s_2)$?

Q6: Impact of $W(s) = 0$ due to $\mu^*(s) = 0$, no loss is assigned to errors for which $\mu^*(s) = 0$ (Eq 7)?

Q7: Eq 8 and Alg 3 (line 19), what if $\mu^*(s) = 0$, and/or $\mu^*_{min} = 0$ where $log$ is undefined?

Q8: Figure 3, No runtimes are reported for WMB. Are these generally similar, a lot lower, or a lot higher than DBE/NeuroBE?

Q9: In Section 5, 'this shows that the algorithm has an anytime characteristic'. Is this about the fact that the local error estimate could be flexibly used to decide on including more training data on the fly? Because I do not understand how the complete NeuroBE algorithm could be considered an anytime algorithm: stopping it at any time will not provide an estimate for $Z$?

Q10: To deal with determinism, the loss function was changed to include a '0'-prediction-term in the loss. What about exactly storing which $s$ configurations have a $0$ probability, for example using a BDD representation. And then only consider the NN output to approximate the other values. Would this be feasible, any thoughts?


# Comments

C1: It would be fair to briefly state at the background beginning that parts of it are copied from the DBE paper.

C2: The experiment Section explains a change to the loss function in order to deal with determinism (0 probability). This should be in Section 3 on the loss function and should not be treated as an afterthought only to mention in the experiments.

C3: The paper uses many symbols making it harder to follow. Especially since several symbols appear to be overloaded, or incorrectly used (unless I'm confused, which is not good either). An example of a seemingly overloaded symbol is $D$ which appears as a set of domains (background), a dataset batch $D$ from a dataset $D_T$ which was in turn created using algorithm 2 outputting $D$, and in 2.2 an example has $D$ as a bucket. An example of a seemingly incorrectly used symbol: Section 3 mentions $\lambda$ will be used for the message computed by exact BE and $\mu^*$ for the locally exact message that potentially uses approximate messages from earlier buckets. But in 2.2 DBE generates a dataset based on $\lambda(S)$ rather than on $\mu^*(S)$? Another example, above and below Theorem 1, $E$ has a different definition?

## Textual remarks:

* "using a NN (line 10)" -> line 9

* "NN-train method in Algorithm 3 (line 5)" -> wrong line?

* "line 14 calculates the partition.." -> line 13

* "each bucket, if it's width.." -> its width?

* In Algorithm 1 and background $F$ is bold. In Algorithm 2 it isn't. Although F in Alg 2 might be a different F than the one in Alg 1, in which case see my previous comment (C3) on overloading symbols.

* Algorithm 2, S appears both bold (domain($S$)) and normal.

* Eq 2 uses $[(b * w)]$ while Eq 3 only uses $(b * w)$ which seems better.

* Eq 6: since D will be a minibatch $D_i$ (Alg 3), perhaps it's best to replace $D$ with $D_i$? "Namely, given a dataset $D_i$,..."

* Figure 3: 'k' is used for both domain size and $10^3$.

* Paragraph on Algorithm 3 references some wrong line numbers.

* Alg 3: Output contains estimated average bucket error; line 20 does not.

* Figure 3: for some 'time' headers, (h) was left out.

* Figure 4a) reports 'k' while b,c,d report 'id' instead

--------

# After rebuttal:

- The comment in the paper on 'anytime characteristic' should be clarified as NeuroBE is, as of now, not an anytime algorithm.

- The runtime of WMB is much faster. This should be reported in the paper as to clearly explain both the current advantage and disadvantage of WMB vs the NN-approximated research direction.

- The authors have expressed willingness to take into account my comments (C1-C3).

I will slightly increase my overall score to Accept.

**Q7 Justification For Your Score:**

Although I consider this to be incremental work, the work itself appears to be well done: proposing a principled way of adapting the NN architecture and training dataset size. For this, the authors analysed the NN complexity to automatically adapt the training dataset accordingly, and analysed the global/local errors and derived bounds (although I did not check the proof in the appendix carefully). I decided on score 6, assuming that the paper will be revised given my comments, especially C3.

**Q9 Complying With Reviewing Instructions:**

1: Yes.

---

### Official Review · Reviewer_GBKn · 2022-04-11

**Q2(1) Originality/Novelty:** 2
**Q2(2) Significance/Impact:** 2
**Q2(3) Correctness/Technical Quality:** 3
**Q2(6) Clarity Of Writing:** 3
**Q6 Overall Score:** 6
**Q8 Confidence In Your Score:** 3

**Q1 Summary And Contributions:**

This paper describes recent work on neural network approximation of bucket elimination. It describes a re-design and improvement of a deep bucket elimination approach published last year.

**Q2 Assessment Of The Paper:**

More detailed information regarding each of these aspects is given below:

**Q2(4) Quality Of Experiments (Optional):**

3: Good: The experimental evaluation is adequate, and the results convincingly support the main claims.

**Q2(5) Reproducibility:**

3: Good: Key resources (e.g., proofs, code, data) are available and key details (e.g., proofs, experimental setup) are sufficiently well-described for competent researchers to confidently reproduce the main results.

**Q3 Main Strengths:**

Using recent advances in neural networks for approximate probabilistic inference is a very promising direction, and there’s value in seeing how far it may lead.

**Q4 Main Weakness:**

It is a promising approach, but also an obvious one. The approach described in this paper is a rather incremental advance over previous work on this topic. Given its focus on how to improve the neural network approximation, and not much else, the paper is not especially interesting to read.

**Q5 Detailed Comments To The Authors:**

I wonder why the paper focuses on the problem of computing the partition function for a Markov network? The bucket elimination approach can be applied to a wide range of problems, and I would have found interesting some discussion of how a similar approach would apply to other problems. Computing the partition function does not require maximization, for example, while other problems of probabilistic inference do.

The amount of detail in the tables is probably necessary, but it is also overwhelming and makes the tables difficult to interpret. I didn't find any explanation for the use of color in the tables.

I suggest not using the abbreviation NN in the title.

**Q7 Justification For Your Score:**

Progress report on using neural network approximation to scale up bucket elimination, especially for computation of the partition function.

**Q9 Complying With Reviewing Instructions:**

1: Yes.

---

### Official Review · Reviewer_uYmR · 2022-04-13

**Q2(1) Originality/Novelty:** 3
**Q2(2) Significance/Impact:** 3
**Q2(3) Correctness/Technical Quality:** 3
**Q2(6) Clarity Of Writing:** 3
**Q6 Overall Score:** 7
**Q8 Confidence In Your Score:** 3

**Q1 Summary And Contributions:**

The paper presents an approach to perform bucket elimination using neural nets. The main contribution is the dynamic NN architecture used compute approximate messages in a scalable manner.


**Q2 Assessment Of The Paper:**

More detailed information regarding each of these aspects is given below:

**Q2(4) Quality Of Experiments (Optional):**

3: Good: The experimental evaluation is adequate, and the results convincingly support the main claims.

**Q2(5) Reproducibility:**

3: Good: Key resources (e.g., proofs, code, data) are available and key details (e.g., proofs, experimental setup) are sufficiently well-described for competent researchers to confidently reproduce the main results.

**Q3 Main Strengths:**

The paper is clearly motivated and has a nice use of neural nets for message passing.
The connection between dynamic NN architecture and messages seems interesting.
The anytime behavior of the method shown in the experiments is also a plus.

**Q4 Main Weakness:**

The approach is based on a recently proposed method, so maybe novelty is not as high though the proposed approach seems sound and works well.

**Q5 Detailed Comments To The Authors:**

The idea is to approximate intractable messages  in bucket elimination due to large widths using neural networks. The approach builds on an existing approach DBE but uses a dynamic NN architecture based on the messages. An importance function is used to weight samples in the loss function of the neural net. Experiments are performed on several benchmarks from UAI and results show gainsaying this approach compared to regular bucket elimination and DBE. Error analysis is performed to bound the results of the approximate messages computed by the NN.

The paper seems well motivated and well written. The choices made are justified nicely and it should be applicable to a large class of problems in graphical models considering the generality of bucket elimination. I liked the fact that the architecture is based on the messages, i.e., it may not require as much tuning as regular type of neural network based methods do. Experimental results show that the improvements of the proposed approach clearly on UAI benchmarks. One question si perhaps why the specific benchmarks were chosen since there are several other UAI benchmarks as well, i.e., is there some limiting factor in applications to any graphical model.

**Q7 Justification For Your Score:**

For me the big plus was the automated choice of architecture that is motivated with justified reasoning of the message width, The work seems to cover all aspects, maybe the experiments can mention why some types of benchmarks were left out or any limitations of the proposed work.

**Q9 Complying With Reviewing Instructions:**

1: Yes.

---

### Official Review · Reviewer_yiRv · 2022-04-14

**Q2(1) Originality/Novelty:** 2
**Q2(2) Significance/Impact:** 3
**Q2(3) Correctness/Technical Quality:** 3
**Q2(6) Clarity Of Writing:** 3
**Q6 Overall Score:** 5
**Q8 Confidence In Your Score:** 3

**Q1 Summary And Contributions:**

The paper considers the problem of estimating the partition function in a given graphical model. Since exact message-passing is computationally intractable, the authors propose NeuralBE, a generalisation of Deep Bucket Elimination (DBE) where the number of hidden units in the NNs depends on the induced width of a bucket function. Since some of the bucket functions will be much simpler, this can translate to savings in the size of their approximations.

**Q2 Assessment Of The Paper:**

More detailed information regarding each of these aspects is given below:

**Q2(4) Quality Of Experiments (Optional):**

3: Good: The experimental evaluation is adequate, and the results convincingly support the main claims.

**Q2(5) Reproducibility:**

3: Good: Key resources (e.g., proofs, code, data) are available and key details (e.g., proofs, experimental setup) are sufficiently well-described for competent researchers to confidently reproduce the main results.

**Q3 Main Strengths:**

- The authors provide a solid argument about how NeuroBE is reducing time/space complexity
- I appreciated how quantities like the sample complexity, etc., were connected to the induced width, e.g., Eq. (2) and (3).
- Sufficient amount of detail in the experimental setup.
- Good that the paper looked into architecture size in the experiments.


**Q4 Main Weakness:**

- Eq (3) connects the sample complexity with the induced width. However, the authors introduce the \eta parameter which is use to calibrate N(w) as "the sample size N(w) often exceeds memory limits for higher width buckets". Could you clarify why it makes sense to use N(w) in the first place if you are modifying its value -by picking an \eta to be small- when we think N(w) is too large?



**Q5 Detailed Comments To The Authors:**

- Though I appreciate the wealth of data in the tables, I found them a bit hard to read, there are too many hues.
- Theorem 1 references a "bucket-chain", but there is no explanation anywhere on the paper what that is.
- The error bound in Theorem 1. uses the notation \eps_{c+k}, but that notation is not explained close to the theorem. The explanation is in the appendix.
- Typography: "NeuroBE" looks like it lives in math mode, i.e., $NeuroBE$. Would be better composed as $\text{NeuroBE}$.


**Q7 Justification For Your Score:**

I would like more clarification around the extra tuning parameter but overall found the paper well written and the contribution to be noteworthy.

**Q9 Complying With Reviewing Instructions:**

1: Yes.

---

### Decision · Program_Chairs · 2022-05-15

**Decision:**

Accept (Poster)

**Comment:**

Meta Review: All reviewers generally appreciated the paper.  There is some room for improvement in the presentation, with feedback provided by the reviewers.